# RRM2 Regulates Hepatocellular Carcinoma Progression Through Activation of TGF-β/Smad Signaling and Hepatitis B Virus Transcription

**DOI:** 10.3390/genes15121575

**Published:** 2024-12-06

**Authors:** Dandan Wu, Xinning Sun, Xin Li, Zongchao Zuo, Dong Yan, Wu Yin

**Affiliations:** 1State Key Lab of Pharmaceutical Biotechnology (SKLPB), College of Life Sciences in Nanjing University (Xianlin Campus), Nanjing University, Nanjing 210046, China; bbmcwdd@126.com (D.W.); sunxinning87@163.com (X.S.); 15248161410@163.com (X.L.); 2The First Affiliated Hospital of Bengbu Medical University, Bengbu 233004, China; zzcmin14@163.com; 3Department of Cardiology, Affiliated Hospital of Nanjing University of TCM, Nanjing 210023, China; y12d12@163.com

**Keywords:** hepatocellular carcinoma, RRM2, TGF-β, Smad2/3, HBV

## Abstract

Background: Hepatocellular carcinoma (HCC) is a type of malignant tumor with high morbidity and mortality. Untimely treatment and high recurrence are currently the major challenges for HCC. The identification of potential targets of HCC progression is crucial for the development of new therapeutic strategies. Methods: Bioinformatics analyses have been employed to discover genes that are differentially expressed in clinical cases of HCC. A variety of pharmacological methods, such as MTT, colony formation, EdU, Western blotting, Q-PCR, wound healing, Transwell, cytoskeleton F-actin filaments, immunohistochemistry (IHC), hematoxylin–eosin (HE) staining, and dual-luciferase reporter assay analyses, were utilized to study the pharmacological effects and potential mechanisms of ribonucleotide reductase regulatory subunit M2 (RRM2) in HCC. Results: RRM2 expression is significantly elevated in HCC, which is well correlated with poor clinical outcomes. Both in vitro and in vivo experiments demonstrated that RRM2 promoted HCC cell growth and metastasis. Mechanistically, RRM2 modulates the EMT phenotype of HCC, and further studies have shown that RRM2 facilitates the activation of the TGF-β/Smad signaling pathway. SB431542, an inhibitor of TGF-β signaling, significantly inhibited RRM2-induced cell migration. Furthermore, RRM2 expression was correlated with diminished survival in HBV-associated HCC patients. RRM2 knockdown decreased the levels of HBV RNA, pgRNA, cccDNA, and HBV DNA in HepG2.2.15 cells exhibiting sustained HBV infection, while RRM2 knockdown inhibited the activity of the HBV Cp, Xp, and SpI promoters. Conclusion: RRM2 is involved in the progression of HCC by activating the TGF-β/Smad signaling pathway. RRM2 increases HBV transcription in HBV-expressing HCC cells. Targeting RRM2 may be of potential value in the treatment of HCC.

## 1. Introduction

Hepatocellular carcinoma (HCC) is the predominant primary liver malignancy, distinguished by significant morbidity and fatality rates [1]. Based on data, there were as many as 906,000 patients with liver cancer diagnosed globally in 2020, and liver cancer has also become the fourth-largest contributor to cancer-related fatalities globally [2]. Studies have shown that approximately 75 to 90 percent of HCC cases develop from cirrhosis caused by persistent infections with viruses (HBV and HCV), alcohol-related damage, and obesity and, to a lesser extent, by genetically predisposed conditions [3]. GLOBOCAN research indicates that HBV is the predominant factor leading to liver cancer, accounting for 56% of all global cases [4]. Approximately two-thirds of liver cancer cases in less developed nations are attributable to HBV, in contrast to one-quarter in more affluent nations. Currently, the treatment of HCC in clinical practice mainly includes local ablative therapy, surgical resection, TACE, radiation therapy, liver transplantation, and systemic drugs [5,6]. Although the therapeutic strategies have improved, the efficacy of treatment for HCC is constrained by drug tolerance, recurrence, and metastasis, leading to a dismal prognosis and serving as the principal cause of mortality among this patient population [6,7]. Therefore, exploring new therapeutic targets and molecular mechanisms is crucial in providing clinical direction in the treatment and prognosis of HCC.

Ribonucleotide reductase (RNR) facilitates the reduction of NDP to dNDP, constituting the critical speed-limiting step in the de novo production of dNTPs [8], which primarily influence genomic stability and cellular proliferation [9]. The RNR comprises two subunits (RRM1 and RRM2) that perform catalytic activity as a tetramer [10]. Recent studies have demonstrated that the dysregulation of RRM2 is associated with certain types of malignancy, including renal cell carcinoma [11], lung cancer [12,13], glioblastoma [14], and breast cancer [15]. In renal cell carcinoma, RRM2 silencing can increase renal cell carcinoma’s sensitivity to sunitinib, as well as improving the anti-tumor efficacy of PD-L1 blockade [11]. Jiang et al. demonstrated that RRM2 regulates lung adenocarcinoma progression by activating cGAS/STING signaling [16]. In addition, Kitab et al. found that RRM2 expression was elevated in HCV-infected liver cells [17], hence augmenting the likelihood of developing HCC. By increasing the protein stability of HCV polymerase NS5B, RRM2 is proposed to be an essential proviral factor for HCV RNA synthesis [8]. In summary, these findings provide strong evidence that RRM2 is involved in the progression of cancer. Nevertheless, RRM2’s specific function in HCC remains uncertain.

Transforming growth factor β (TGF-β) is a prototype of the TGF-β superfamily, which is composed of TGF-β, activins, inhibins, bone morphogenetic proteins (BMPs), and growth and differentiation factors (GDFs) [18,19]. As a pleiotropic cytokine, TGF-β (TGF-β1, TGF-β2, and TGF-β3) plays an important role in early embryonic development and adult homeostasis [19]. Increasing evidence indicates that TGF-β signaling is actively involved in tumor progression, metastasis, immune response suppression, and vascular remodeling [20]. Recent findings show that TGF-β can induce epithelial-to-mesenchymal transition (EMT), and the elevated expression of TGF-β was correlated with poor overall survival (OS) in esophageal adenocarcinoma [21].

In the current study, we discovered that RRM2 was elevated in HCC, and elevated RRM2 expression was correlated with a dismal prognosis in HCC patients. Furthermore, silencing RRM2 expression in HCC significantly attenuated cell proliferation and metastasis. Both bioinformatics analysis and functional experiments revealed that the pro-tumorigenic effect of RRM2 on HCC is closely related to the activation of TGF-β/Smad signaling. In addition, RRM2 increases HBV transcription in HBV-expressing HCC cells.

## 2. Materials and Methods

### 2.1. Materials

jetPRIME^®^ was obtained from Polyplus (Strasbourg, France). The following antibodies were used: Tubulin (Cat#BS1699), Flag (Cat#AP0007), and TGF-β1 (Cat#BS1361) were obtained from Biogot (Nanjing, China); N-cadherin (Cat#22018-1-AP), Vimentin (Cat#60330-1-Ig), PCNA (Cat#24036-1-AP), and E-cadherin (Cat#60335-1-Ig) were procured from Proteintech (Wuhan, China); Smad2 (Cat#ET1604-22), Smad3 (Cat#ET1607-41), p-Smad2 (Cat#ET1702-34), and p-Smad3 (Cat#ET1609-41) were obtained from Huabio (Hangzhou, China); RRM2 (Cat#sc-398294) was purchased from Santa Cruz (Starr, TX, USA). The HCC tissue microarray (TMA, Cat. HLiv030PG-1) was obtained from Outdo Biotech (Shanghai, China). Plasmids of pGL3 basic, Cp, Xp, spI, spII, PRL, and HBV were obtained from Miaoling Biotech (Wuhan, China). The RRM2-overexpressing plasmid (Flag-RRM2) was purchased from General Biosystems (Chuzhou, China).

### 2.2. Cell Lines and Cell Culture

The hepatocellular carcinoma cell lines SMMC7721, Huh7, H22, HepG2, and HepG2.2.15, and the immortalized hepatocyte cell line LO2, were purchased from the Shanghai Cell Bank (Shanghai, China). Cells were maintained and cultured in DMEM (WISENT, Saint-Jean-Baptiste, QC, Canada) medium containing 1% penicillin–streptomycin (WISENT, Saint-Jean-Baptiste, QC, Canada) and 10% fetal bovine serum (FBS) (ExCell, Shanghai, China). All cell lines were cultured at 37 °C with 5% CO_2_.

### 2.3. Immunohistochemistry (IHC) Staining

Paraffin sections of normal or tumor tissue were dewaxed with xylene (Sinopharm, Nanjing, China) and moved to a continuously diluted ethanol solution (Aladdin, Shanghai, China), and the sections were subjected to antigen repair with citric acid solution at approximately 95 °C for 20 min. We sealed the sections with 5% BSA for 30 min. The primary antibody was incubated overnight at 4 °C and the secondary antibody for 1 h. The signal was identified utilizing the DAB Kit (Servicebio, Wuhan, China). Photos were taken utilizing an optical microscope (Nexcope, Ningbo, China).

### 2.4. Cell Transfection

Flag-RRM2 and siRNAs were transfected into cells using jetPRIME^®^ (Polyplus, Strasbourg, France), following the instructions of the manufacturer. We diluted the siRNA in jetPRIME^®^ buffer, mixed it well, added the jetPRIME^®^ reagent, and incubated it at room temperature for 10 min. Finally, we added the transfection mixture to the cells containing the culture medium and continued to incubate the cells for 24 to 72 h. In this study, the following siRNAs were used: RRM2 siRNA#1: 5′-GGAGAGUAAGAAAUATT-3′, RRM2 siRNA#2: 5′-GAGCCGCUGCUGAGAGAAATT-3′. The control siRNA was 5′-UUCUCCGAACGUGUCACGUTT-3′.

### 2.5. Real-Time Quantitative PCR (Q-PCR)

Cells were harvested; RNA was obtained utilizing a total RNA extraction reagent; and the reverse transcription of the RNA, followed by Q-PCR analysis, was conducted utilizing the ABScript III RT Master Mix and a 2X Universal SYBR Green Fast qPCR Mix (ABclonal, Wuhan, China), in accordance with the producer’s guidelines. Data were collected by a fluorescent quantitative PCR instrument (Steponeplus, ABI, Waltham, MA, USA). The exact sequences of the Q-PCR primers utilized in this work are shown in Appendix A.

### 2.6. Western Blotting

Cell or tissue proteins were extracted, and then the protein concentration was quantified by the BCA method. Proteins were separated by SDS-PAGE and subsequently transferred to PVDF membranes (Millipore, Bedford, MA, USA). The PVDF membranes were blocked with 5% milk for 1 h and incubated with the specific primary antibodies at 4 °C overnight. Then, the membranes were incubated with horseradish peroxidase-conjugated secondary antibodies for 1 h. Finally, the membrane was imaged in a gel imager using the ECL kit (FDbio, Hangzhou, China).

### 2.7. Cell Viability Assays

Cells were diluted with media to a concentration of 5 × 10^4^ cells/mL and subsequently injected into a 96-well cell culture plate (100 μL/well). Following treatment, 20 μL of MTT working solution was added per well, and the cells were cultivated for an additional 2–4 h. Then, DMSO (150 μL/well) was added and allowed to react for 30 min in the absence of light. For the purpose of analysis, a spectrophotometer (ELx800TM, Bio-Tek, Shoreline, WA, USA) was utilized to measure the optical density at a wavelength of 490 nm.

### 2.8. Colony-Forming Assay

Cells were diluted with media to a concentration of 5 × 10^3^ cells/mL and subsequently injected into a 3.5 cm cell culture dish. The cells were treated and grown for 7–15 days to form colonies; then, they were fixed with 4% PFA fix solution (Beyotime, Wuhan, China) and dyed with crystal violet solution (Beyotime, Wuhan, China). Lastly, they were washed with PBS and imaged using a camera.

### 2.9. EdU Assay

In accordance with the producer’s guidelines, the BeyoClick™ EdU cell proliferation kit with Alexa Fluor 594 (Beyotime, Wuhan, China) was utilized in order to determine the cells’ capability for proliferation. Briefly, the treated cells were seeded into 24-well plates (1.5 × 10^5^/well) and incubated with 10 μM EdU at 37 °C for 2 h. The cells were then fixed using 4% paraformaldehyde for 15 min and permeabilized using 0.3% Triton X-100 for 10 min. After washing with PBS, the cells were stained with the Click Additive Solution in the dark for 30 min, and the cell nuclei were stained with DAPI. We captured photos using a fluorescence microscope (Nexcope, Ningbo, China).

### 2.10. Wound Healing Assays

Cells were seeded in 12-well plates (1.5 × 10^5^ cells/mL). After treatment, linear scratch wounds in the cell cultures were prepared using a sterile 100 μL pipette tip. The cells were then cultured in fresh serum-free medium for 36 h. Cell wound healing was photographed with an inverted optical microscope (Nexcope, Ningbo, China) at 0 and 36 h.

### 2.11. Transwell Assay

After the cells were treated, they were diluted with serum-free culture media and inoculated into the upper chamber (2 × 10^4^ cells/mL), and a culture medium with 10% FBS was added to the bottom chamber. After incubation at 37 °C for 24 h, the cells were fixed with 4% paraformaldehyde for 20 min and stained with 1% crystal violet for 15 min. Images were obtained with an inverted light microscope.

### 2.12. Cytoskeleton F-Actin Filament Assay

Cytoskeletal F-actin filaments were detected using iFluor™ 488 phalloidin (green fluorescence) reagent (Yeasen, Shanghai, China), according to the manufacturer’s instructions. Briefly, cells were inoculated into 24-well plates, and, 36 h after transfection, they were fixed with 4% paraformaldehyde for 15 min and permeabilized with 0.1% Triton X-100 for 10 min. After washing with PBS, the cells were stained with freshly prepared iFluor™ 488 phalloidin working solution for 60 min in the dark, and then the nuclei were stained with DAPI. Photographs were taken using a fluorescence microscope.

### 2.13. Establishment of the Mouse Tumor Model

SPF male C57BL/6 mice weighing 18–22 g were purchased from GemPharmatech (Nanjing, China) for the purpose of HCC modeling. H22 cells were administered into the peritoneal cavities of the mice, which were subsequently maintained on a feeding regimen for approximately one week. Cells were taken from the abdominal cavities of the mice, and the final cell concentration was diluted with saline to 2 × 10^7^ cells/mL. To establish an orthotopic tumor-bearing mouse model, 200 μL of the diluted cell combination was administered into the liver tissue, whereas the control group received a saline injection. The orthotopic tumor mice were randomized into three groups (*n* = 6) and subsequently injected with an adeno-associated virus (AAV)-expressing control (AAV-shNC), AAV-expressing shRRM2 (AAV-shRRM2), or normal saline (control) through the tail vein. After treatment, the mice were maintained on a standard diet for 2 weeks. Subsequent to the experiment, the blood, livers, and lungs of the mice were harvested, and the animals were euthanized by cervical dislocation.

### 2.14. Hematoxylin–Eosin (HE) Staining

Xylene was employed to deparaffinize the tissue sections. The xylene was removed using serial dilutions of ethanol (100%, 95%, and 80%, 5 min each). The sections were stained using HE staining solution (Servicebio, Wuhan, China), dehydrated with ethanol solutions with increasing concentrations, and mounted using neutral resin. Ultimately, pictures were obtained with an optical microscope.

### 2.15. Genomic DNA Extraction

To extract the genomic DNA, a DNA virus genome extraction kit (Solarbio, Beijing, China) was used. The cells’ supernatant was collected and centrifuged, and 500 μL of the supernatant was aspirated. Subsequently, proteinase K (20 μL) was introduced and the cells were treated at 65 °C for 15 min. Solution V (500 μL) was subsequently introduced, and anhydrous ethanol (400 μL) was incorporated after the mixture was thoroughly mixed. The mixture was eluted using an adsorption column according to the manufacturer’s instructions.

### 2.16. Dual-Luciferase Reporter Assay

The pRL-TK plasmid and transfection reagent were co-transfected into cells, following the manufacturer’s guidelines. Following treatment, the cells were collected, and the promoter activity of the treated cells was assessed utilizing a chemiluminescence detector (Promega, Madison, WI, USA). Eventually, the firefly to Renilla luciferase activity ratio was recorded and analyzed.

### 2.17. Statistical Analysis

Statistical analysis was performed using Prism 9 (GraphPad, La Jolla, CA, USA). Data were expressed as the mean ± SEM, and the experiments were conducted at least three times independently. Student’s *t*-test was utilized to evaluate the differences between two groups, while an ANOVA was utilized to evaluate the differences across several groups. *p* < 0.05 indicated a statistically significant difference, whereas ns signified no statistical significance.

## 3. Results

### 3.1. RRM2 Is Highly Expressed in HCC and Predicts Poor Prognosis

An RNA sequence analysis was previously carried out by our research team on three pairs of clinical HCC and surrounding noncancerous tissue samples in order to examine genes that are expressed differently in HCC [22]. From the results of the RNA-seq study, it was discovered that RRM2 was considerably upregulated in HCC tissue in comparison to surrounding noncancerous tissue, with |log2FC| > 2 and *p*-value < 0.001 [22]. Meanwhile, a TIMER database analysis revealed that, in various types of cancer, including HCC, the expression of RRM2 was specifically elevated in tumor tissue compared to adjacent noncancerous tissue (Figure 1A). Similar results were also identified in the analysis of the GEPIA database (Figure 1B). To further validate the findings, note that the RRM2 expression was markedly elevated in HCC tissue relative to the surrounding noncancerous tissue (Figure 1C). Furthermore, a Kaplan–Meier analysis showed that elevated expression levels of RRM2 were correlated with diminished survival in HCC patients exhibiting varying degrees of differentiation (Figure 1D), alongside reduced progression-free survival (PFS), recurrence-free survival (RFS), and overall survival (OS) (Figure 1E).

### 3.2. RRM2 Gene Contributes to the Growth of HCC

To further understand the significance of RRM2 in HCC, we looked at the RRM2 expression in HCC cell lines such as HepG2.2.15, HepG2, SMMC7721, Huh7, and LO2 cells (immortalized hepatocyte cell line). Both Western blotting and Q-PCR analyses indicated that, in comparison to LO2 cells, the protein and mRNA levels of RRM2 were elevated in HCC cells (Figure 2A,B). In SMMC7721 and Huh7 cells, transfection with RRM2 siRNA (si-RRM2#1 and siRRM2#2) significantly decreased the RRM2 protein and mRNA levels (Figure 2C). Subsequently, we utilized MTT, colony formation, and EdU experiments in order to determine the impact of RRM2 knockdown on the proliferation of SMMC7721 and Huh7 cells. The results demonstrated that the suppression of RRM2 substantially impeded cell proliferation (Figure 2D,F). Conversely, the overexpression of RRM2 in SMMC7721 and Huh7 cells resulted in a remarkable enhancement in cell proliferation in comparison to control cells (Figure 2G–J).

### 3.3. RRM2 Regulates the EMT Phenotype in HCC

To explore whether RRM2 affects HCC cell migration, we utilized wound healing and Transwell experiments to evaluate SMMC7721 and Huh7 cell migration with knocked down or overexpressed RRM2. The experimental findings demonstrated that the migration ability in HCC cells was significantly inhibited when RRM2 was knocked down, whereas the overexpression of RRM2 promoted HCC cell migration (Figure 3A–D). EMT is a biological process in which cancer cells lose their epithelial-like phenotype and transform into a spindle-shaped mesenchymal phenotype. It is a key indicator of tumor cell metastasis and plays an important role in the malignant transformation of cancer cells. In the present study, we observed that RRM2 overexpression in SMMC7721 and Huh7 cells induced alterations in the cell shape, resembling an EMT phenotype, whereas RRM2 knockdown had the opposite effect (Appendix A). Meanwhile, RRM2 knockdown significantly reduced F-actin formation (Figure 3E), whereas forced RRM2 expression promoted F-actin overgrowth in SMMC7721 and Huh7 cells (Figure 3F), indicating that RRM2 plays an important role in cytoskeletal reorganization and facilitates tumor cell migration. Furthermore, RRM2 silencing resulted in increased E-cadherin expression but inhibited the expression of N-cadherin and vimentin at both the protein and mRNA levels in SMMC7721 and Huh7 cells (Figure 3G,H). Conversely, RRM2 overexpression had the opposite effect (Figure 3I,J).

### 3.4. RRM2 Regulates HCC Migration Through the TGF-β/Smad Signaling Pathway

TGF-β is well known to be a critical factor that induces EMT in cells [23]. TGF-β has a variety of biological functions that regulate cell differentiation, apoptosis, proliferation, and migration [24,25,26]. TGF-β is able to bind to its receptors and activate downstream Smad proteins (Smad2 or Smad3) via phosphorylation and nuclear translocation [27], and studies have indicated that the TGF-β/Smad pathway is crucial in controlling HCC growth and progression [28]. To investigate whether RRM2 had a relationship with the TGF-β/Smad pathway in HCC, we performed a GEPIA2 database analysis. The results showed that RRM2 expression was related to the expression of TGFB1, Smad2, and Smad3 (Figure 4A). Meanwhile, the TGF-β1 mRNA levels were significantly reduced when RRM2 was silenced in SMMC7721 and Huh7 cells (Figure 4B). In SMMC7721 and Huh7 cells, the knockdown of RRM2 reduced the TGF-β1 protein expression, as well as the phosphorylation of the Smad2 and Smad3 proteins (Figure 4C), whereas the overexpression of RRM2 produced the opposite effect (Figure 4D,E). Furthermore, the wound healing and Transwell assays revealed that RRM2 silencing impaired TGF-β-induced cell migration (Figure 4F,G). SB431542 is an inhibitor of TGF-β signaling. The wound healing and Transwell assays revealed that SB431542 markedly suppressed RRM2-induced cell migration (Figure 4H,I). Taken together, RRM2 overexpression enhances HCC cell migration through the TGF-β/Smad signaling pathway.

### 3.5. RRM2 Presents Proto-Oncogene Activity in a Mouse Tumor Model of HCC

The mouse tumor model of HCC was established by the intraperitoneal injection of H22 cells to assess the tumorigenic effect of RRM2 in vivo [22]. RRM2 expression in mouse liver tissue was suppressed by adeno-associated virus-mediated shRNA specifically targeting RRM2 (AAV-shRRM2). Q-PCR analysis demonstrated that RRM2 expression was markedly elevated in H22 tumors as compared to normal liver tissue, and AAV-shRRM2 delivery successfully reduced the RRM2 expression in the mouse liver (Figure 5A). Western blotting analysis demonstrated that AAV-shRRM2 delivery resulted in the decreased expression of RRM2 and PCNA in the mouse liver (Figure 5B). Meanwhile, we observed a substantial decrease in tumor size in the AAV-shRRM2 group when compared to the AAV-shNC group (Figure 5C), and AAV-shRRM2 delivery also led to a decrease in the liver weight (Figure 5D). Additionally, RRM2 silencing was related to a decrease in serum ALT and AST levels (Figure 5E). The HE analysis of the liver tissue revealed that the tumor lesions were reduced in AAV-shRRM2 mice (Figure 5F). Notably, pulmonary metastatic nodules were observed in both the AAV-shNC and AAV-shRRM2 groups, indicating the occurrence of lung metastasis in H22 tumors. However, compared to the control group, RRM2-silenced mice exhibited significantly fewer metastatic nodules (Figure 5G,H). Furthermore, IHC analysis confirmed that RRM2 knockdown decreased the protein expression of PCNA, p-Smad2, and p-Smad3 (Figure 5I).

### 3.6. RRM2 Regulates HBV Transcription

HBV infection is a primary etiological factor of HCC, so it is crucial to investigate potential mechanisms that modulate HBV [29]. Because RRM2 is reported to induce viral replication, we examined whether it influences HBV replication. The Kaplan–Meier analysis indicated that elevated RRM2 expression was correlated with diminished survival in HBV-associated HCC patients (Figure 6A). Consequently, we hypothesized that RRM2 served a significant regulatory function in HBV-associated HCC. HBV pgRNA has emerged as an HBV serological marker. The original template for pgRNA is HBV cccDNA. We selected HBV RNA, pgRNA, cccDNA, and HBV DNA for analysis as indicators of HBV replication. In this study, we found that the mRNA levels of HBV RNA, pgRNA, cccDNA, and HBV DNA were all reduced in RRM2-silenced HepG2.2.15 cells with stable HBV infection (Figure 6B). Next, the mRNA levels of HBV RNA, pgRNA, cccDNA, and HBV DNA were markedly increased in SMMC7721 and HepG2 cells transfected with an HBV plasmid (Figure 6C). Under this circumstance, the mRNA level of RRM2 was also remarkably increased by HBV plasmid transfection (Figure 6D). The HBV genome has four promoters, the preS2 promoter (SpII), the preS1 promoter (SpI), the X gene promoter (Xp), and the core promoter (Cp), along with two enhancers. The Cp and Xp promoters exhibited the highest activity. To examine the influence of RRM2 on HBV transcription, four HBV promoter fragments were cloned into the pGL3-basic promoter vector (Figure 6E). The dual-luciferase reporter assay indicated that RRM2 knockdown inhibited the promoter activity of Cp, Xp, and SpI in SMMC7721 and HepG2 cells transfected with the HBV plasmid (Figure 6F). These results indicate that RRM2 knockdown inhibits HBV replication at the transcriptional level.

## 4. Discussion

In this study, we found that RRM2 expression was increased in HCC, which was well correlated with negative clinical outcomes. RRM2 knockdown significantly inhibited HCC cell growth and metastasis. Meanwhile, bioinformatics analysis and functional experiments revealed that RRM2 promoted the malignant progression of HCC, which was highly correlated with TGF-β/Smad signaling activation. Moreover, RRM2 was related to a poor prognosis in HBV-associated HCC patients and influenced HBV replication by regulating HBV promoter activity. The above results suggest that RRM2 is a prospective target for HCC treatment and drug development.

Growing evidence indicates that TGF-β is a potential target for cancer treatment [30], and TGF-β has several biological functions, including the regulation of cell adhesion, differentiation, senescence, and migration [31]. TGF-β can also activate downstream Smad proteins, and the TGF-β/Smad signaling pathway is strongly correlated with tumorigenesis and the progression of various tumors [32]. Among the various subtypes of TGF-β, TGF-β1 exerts its biological effects by activating downstream mediators such as Smad2 and Smad3, whereas its activity is adversely controlled by the expression of Smad7 [33]. TGF-β1 regulates cell migration and EMT by activating Smad2/3 phosphorylation [34]. Recent research shows that RRM2 promotes liver metastasis in pancreatic cancer by stabilizing YBX1 and activating the TGF-β pathway [35]. In the present investigation, we identified a new role of RRM2 in regulating the TGF-β/Samd signaling pathway in HCC. RRM2 promotes HCC cell migration by reducing E-cadherin expression and enhancing the expression of vimentin and N-cadherin. The GEPIA2 database analysis showed that the expression of RRM2 was correlated with the expression of TGFB1, Smad2, and Smad3. Q-PCR and Western blotting experiments showed that RRM2 upregulated the mRNA and protein levels of TGF-β1, as well as the mRNA expression levels of Smad2 and Smad3 (Appendix A), but RRM2 had no significant effect on the protein levels of Smad2 and Smad3. It has been shown that the phosphorylation of Smad2 or Smad3 linkers is ubiquitinated and then degraded by the proteasome [36,37,38]. Therefore, we hypothesized that RRM2 may affect the phosphorylation of the Smad linker, leading to the partial degradation of Smad2 and Smad3 proteins. The above results demonstrate the close connection between RRM2 and TGF-β signaling. SB431542 is recognized as a potent inhibitor of TGF-β signaling, which is achieved by inhibiting TGF-β RI activity [39]. SB431542 treatment blocked RRM2-mediated EMT and cell migration, which further confirms the critical impact of RRM2 on TGF-β signaling. Since RRM2 affects TGF-β signaling, in this study, we also evaluated the changes in the expression of other typical TGF-β target genes, such as PAI-1 and Smad7 [40], when RRM2 was knocked down or overexpressed, and the results showed that RRM2 was able to upregulate the mRNA expression of PAI-1, while it did not have a significant effect on the mRNA expression of Smad7 (Appendix A). In summary, although RRM2 is a small subunit involved in nucleotide metabolism, it regulates HCC by activating the TGF-β/Smad signaling pathway.

HCC ranks as the sixth most prevalent malignant neoplasm globally and seriously endangers human life [41]. This phenomenon is particularly evident in Asia and Africa, primarily due to the fact that 3% to 15% of the population in these regions is affected by chronic HBV [42]. HBV is a liver-transmissible DNA virus characterized by its envelope and capsid, exhibiting strong infectivity [43]. It enters the nucleus through endocytosis, releasing rcDNA to form cccDNA, which is then transcribed into pgRNA [44]. pgRNA can be used as a viral replication template and also circulates with cccDNA to sustain the cccDNA levels in HBV-infected cells [45]. Thus, HBV replication is difficult to eliminate, leading to chronic hepatitis B, and can promote HCC [46,47]. Previous studies have indicated that the HBV infection of cells can directly or indirectly cause the dysregulation of TGF-β expression and activate various oncogenic mechanisms that contribute to the malignant transformation of HCC, including EMT, metastasis, fibrosis, apoptotic processes, proliferation, and inflammatory responses [48]. Yang et al. discovered that increased TGF-β activity, linked to the persistence of HBV in hepatic tissue, suppressed the expression of microRNA-34a, resulting in the heightened production of the chemokine CCL22 and promoting immune evasion [49]. Ye et al. found that the amalgamation of TGF-β shRNA and HBV double shRNA decreased blood and tissue HBV DNA, HBV RNA, and liver fibrosis indicators while enhancing the liver morphology more efficaciously than medication alone [50]. The investigation and identification of targets that can modulate HBV-TGF-β may offer new potential for the treatment of hepatocellular carcinoma and the development of pharmaceuticals. Increasing evidence indicates that RRM2 plays an essential role in the productive replication of multiple viruses, including HCV, HPV, ALV-J, and COVID-19 [8,51,52,53]. In this study, we found that RRM2 not only activates the TGF/Smad signaling pathway but also modulates HBV replication by regulating HBV promoter activity.

## 5. Conclusions

RRM2 is involved in the progression of HCC by activating the TGF-β/Smad signaling pathway. RRM2 increases HBV transcription in HBV-expressing HCC cells. Targeting RRM2 may have potential value in the treatment of HBV-associated HCC.

## Figures and Tables

**Figure 1 genes-15-01575-f001:**
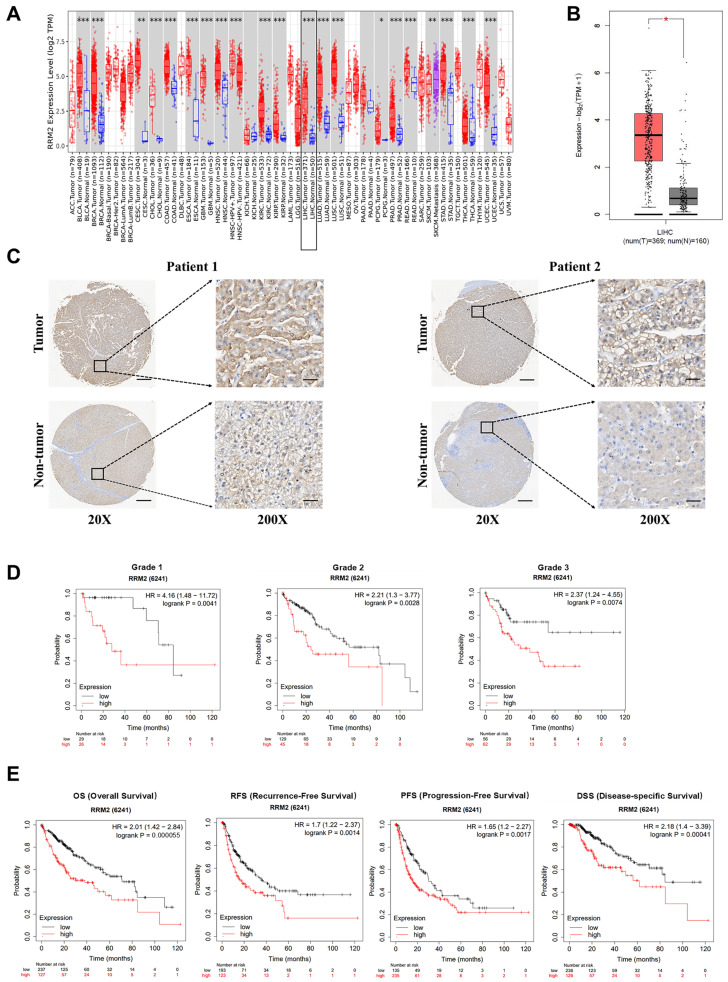
RRM2 upregulation in HCC is associated with poor prognosis. (**A**) RRM2 expression in different types of cancer was analyzed using the TIMER database (“https://cistrome.shinyapps.io/timer/ (accessed on 5 May 2023)”). The black box shows HCC-related data. (**B**) The GEPIA2 database (“http://gepia2.cancer-pku.cn/#index (access date: 5 May 2023)”) was used to analyze the expression levels of RRM2 in HCC tissue (*n* = 369) and precancerous tissue (*n* = 160). (**C**) Representative RRM2 IHC staining images from a clinical TMA of HCC (*n* = 15). Magnification: 20× and 200×. (**D**) Kaplan–Meier analysis was used to examine the relationship between RRM2 expression levels and OS, RFS, and PFS in HCC patients (“http://kmplot.com/analysis/ (accessed on 5 May 2023)”), with a low RRM2 expression group (*n* = 29/129/56) and a high RRM2 expression group (*n* = 26/45/62). (**E**) Kaplan–Meier analysis was utilized to analyze RRM2 expression in HCC patients with variable degrees of differentiation (“http://kmplot.com/analysis/ (accessed on 5 May 2023)”), with a low RRM2 expression group (*n* = 237/193/135) and a high RRM2 expression group (*n* = 127/123/235). * *p* < 0.05, ** *p* < 0.01, and *** *p* < 0.001.

**Figure 2 genes-15-01575-f002:**
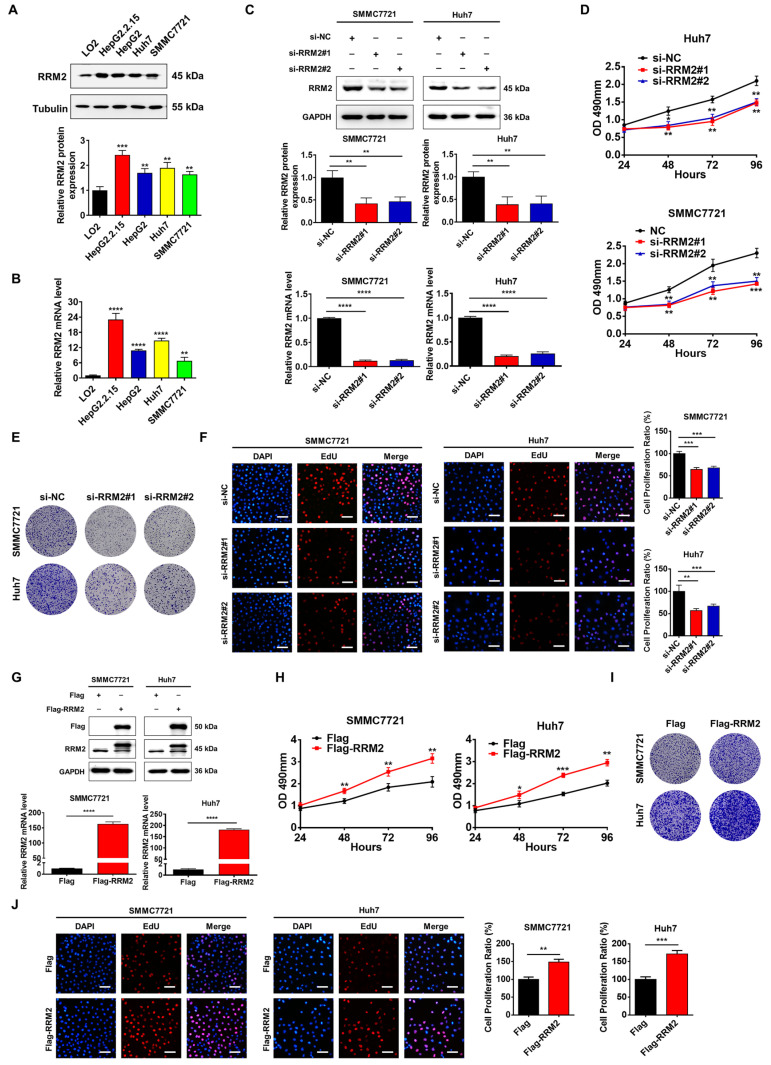
RRM2 expression promotes the proliferation of HCC cells. (**A**,**B**) Western blotting and Q-PCR tests were used to assess the expression of RRM2 in HCC cell lines, such as HepG2.2.15, HepG2, SMMC7721, Huh7, and the immortalized hepatocyte cell line LO2. (**C**) Western blotting and Q-PCR tests were utilized to evaluate the expression of RRM2 in SMMC7721 and Huh7 cells transfected with RRM2 siRNA (siRRM2#1 and siRRM2#2). (**D**) MTT test was applied to evaluate RRM2 siRNA’s impact on SMMC7721 and Huh7 cell viability. (**E**) Colony-forming test was utilized to evaluate RRM2 siRNA’s impact on SMMC7721 and Huh7 cell colony formation. (**F**) Representative EdU staining (red) images of SMMC7721 and Huh7 cells expressing low RRM2 levels. Nuclei are counterstained with DAPI (blue). Scale bar: 200 µm. (**G**) Western blotting and Q-PCR tests were utilized to evaluate the expression of RRM2 in SMMC7721 and Huh7 cells transfected with Flag-RRM2. (**H**) MTT test was utilized to evaluate SMMC7721 and Huh7 cell viability after RRM2 overexpression. (**I**) A colony-forming test was utilized to evaluate SMMC7721 and Huh7 cell colony formation after RRM2 overexpression. (**J**) Representative EdU staining (red) images of SMMC7721 and Huh7 cells expressing high RRM2 levels. Nuclei are counterstained with DAPI (blue). Scale bar: 200 µm. * *p* < 0.05, ** *p* < 0.01, *** *p* < 0.001, and **** *p* < 0.0001.

**Figure 3 genes-15-01575-f003:**
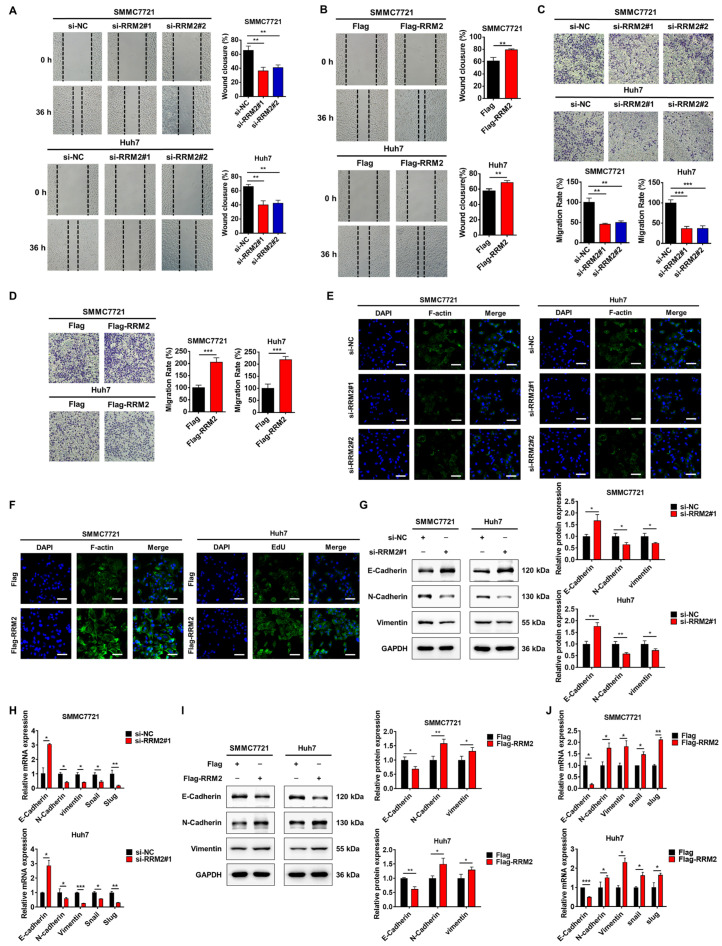
RRM2 promotes HCC cell migration. (**A**–**D**) Wound healing and Transwell assays were performed to assess the migration of SMMC7721 and Huh7 cells after RRM2 knockdown or overexpression. (**E**,**F**) Representative F-actin (green) images of SMMC7721 and Huh7 cells after RRM2 knockdown or overexpression. Nuclei are counterstained with DAPI (blue). Scale bar: 200 µm. (**G**–**J**) Western blot and Q-PCR tests revealed the impact of RRM2 knockdown or overexpression on EMT markers in SMMC7721 and Huh7 cells. * *p* < 0.05, ** *p* < 0.01, and *** *p* < 0.001.

**Figure 4 genes-15-01575-f004:**
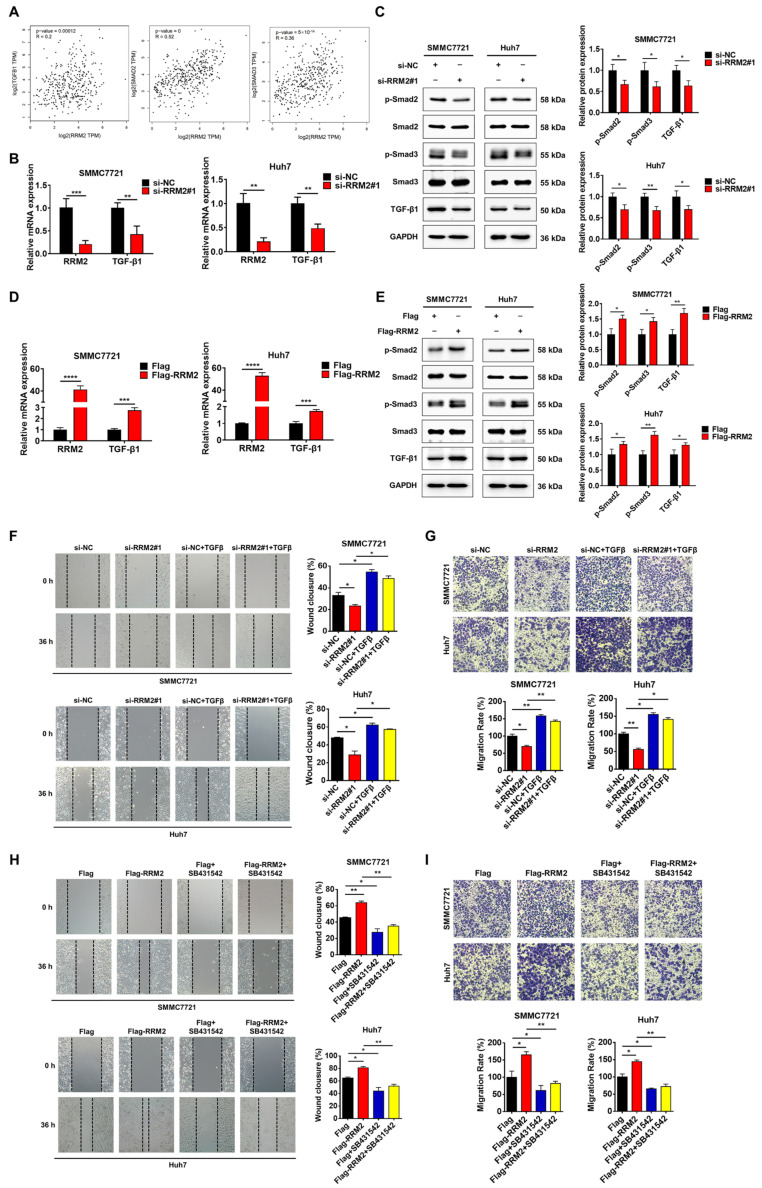
RRM2 promotes HCC migration by activating TGF-β/Smad signaling. (**A**) The GEPIA2 database was analyzed for correlations between RRM2 and TGFB1, Smad2, and Smad3 gene expression (“http://gepia2.cancer-pku.cn/#index (accessed on 19 July 2023)”). (**B**) Q-PCR assay to detect TGF-β1 mRNA levels in SMMC7721 and Huh7 cells after RRM2 knockdown. (**C**) Western blotting assay was utilized to evaluate the protein expression of TGF-β1, p-Smad2, and p-Smad3 in SMMC7721 and Huh7 cells transfected with RRM2 siRNA. (**D**) Q-PCR assay to detect TGF-β1 mRNA levels in SMMC7721 and Huh7 cells after RRM2 overexpression. (**E**) Western blotting assay was utilized to evaluate RRM2 protein expression in SMMC7721 and Huh7 cells transfected with Flag-RRM2. (**F**,**G**) SMMC7721 and Huh7 cells underwent a 24 h preliminary treatment with siRRM2#1, followed by a 12 h treatment with TGF-β1 (5 ng/mL). Transwell and wound healing tests assessed cell migration. (**H**,**I**) SMMC7721 and Huh7 cells underwent a 12 h preliminary treatment with Flag-RRM2, followed by a 24 h treatment with SB431542 (5 μM). SMMC7721 and Huh7 cells were transfected with Flag-RRM2 for 12 h and subsequently treated with SB431542 (5 μM) for 24 h. Transwell and wound healing tests assessed cell migration. * *p* < 0.05, ** *p* < 0.01, *** *p* < 0.001, and **** *p* < 0.0001.

**Figure 5 genes-15-01575-f005:**
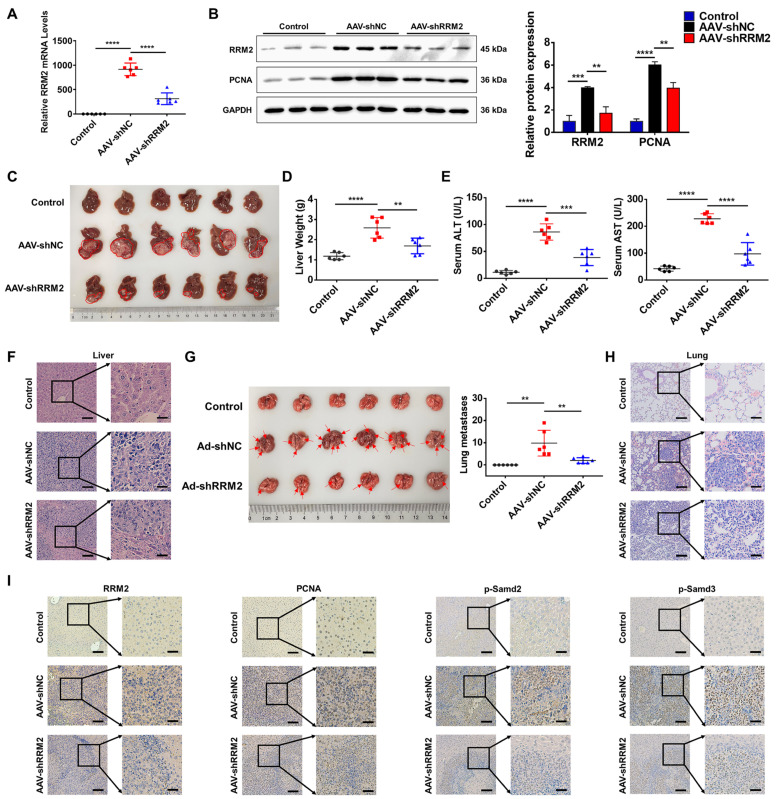
RRM2 knockdown inhibits tumorigenesis in a mouse model of HCC. (**A**) Q-PCR test was utilized to evaluate the RRM2 mRNA levels in mouse liver tissue in the control, AAV-shNC, and AAV-shRRM2 groups (*n* = 6). (**B**) Western blotting analysis was used to evaluate RRM2 and PCNA protein levels in the liver tissue of the control, AAV-shNC, and AAV-shRRM2 groups (*n* = 3). (**C**) Images of the morphology of the liver tissue from the orthotopic transplantation mouse model of HCC are shown (*n* = 6). (**D**) Mouse liver weight in the control, AAV-shNC, and AAV-shRRM2 groups. (**E**) The AST and ALT levels in the serum of the control, AAV-shNC, and AAV-shRRM2 groups (*n* = 6). (**F**) Representative HE images of liver tissue from control, AAV-shNC, and AAV-shRRM2 mice. (**G**) Representative HE images of lung histomorphology and number of lung metastases in orthotopic transplantation tumor mouse models (*n* = 6). (**H**) Representative HE images of lung tissue from control, AAV-shNC, and AAV-shRRM2 mice. (**I**) Representative IHC images of RRM2, p-Smad2, p-Smad3, and PCNA expression in liver tissue of control, AAV-shNC, and AAV-shRRM2 mice. Scale bars: 100 and 35 μm. ** *p* < 0.01, *** *p* < 0.001, and **** *p* < 0.0001.

**Figure 6 genes-15-01575-f006:**
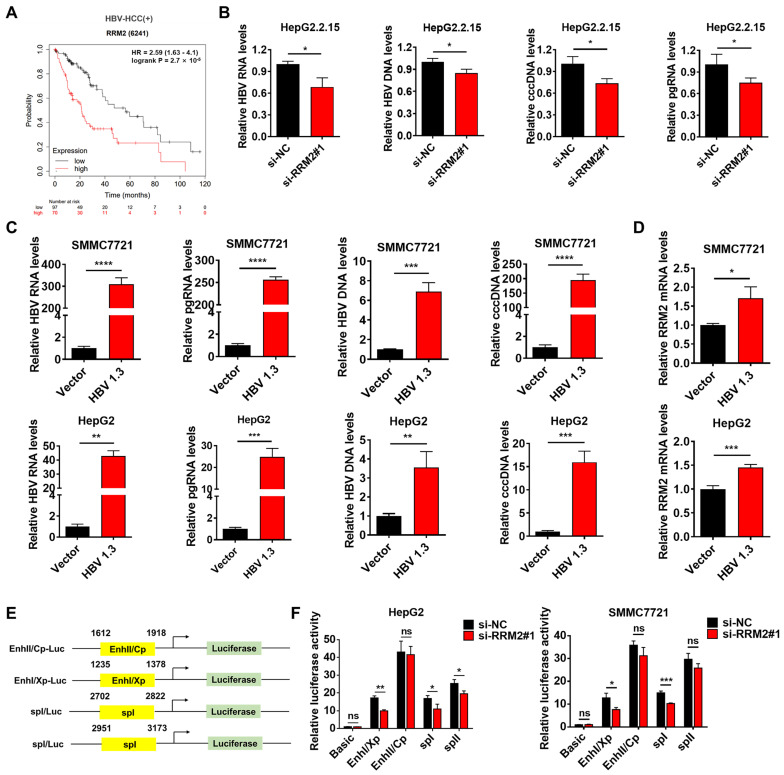
RRM2 promotes HBV replication. (**A**) Kaplan–Meier survival plots of HBV-associated HCC patients with different RRM2 mRNA levels in the low RRM2-expressing group (*n* = 97) and the high RRM2-expressing group (*n* = 70). (**B**) The mRNA levels of pgRNA, HBV DNA, HBV RNA, and cccDNA were detected by Q-PCR in HepG2.2.15 cells transfected with RRM2 siRNA (siRRM2#1). (**C**,**D**) The mRNA levels of RRM2, HBV RNA, pgRNA, HBV DNA, and cccDNA were measured by Q-PCR in SMMC7721 and HepG2 cells transfected with the HBV plasmid. (**E**) Schematic representation of luciferase reporter plasmid containing HBV promoter. (**F**) The HBV plasmid was transfected into SMMC7721 and HepG2 cells and continued to culture for 24 h, and then pGL3 basic, Cp, Xp, spI, and spII were co-transfected with siNC or siRRM2#1 and continued to culture for 36 h. The promoter activity of Cp, Xp, spI, and spII was measured using a dual-luciferase reporter assay. * *p* < 0.05, ** *p* < 0.01, *** *p* < 0.001, **** *p* < 0.0001; ns represents nonsignificant effects.

## Data Availability

The data that support the findings of this study are available from the corresponding author, Wu Yin, upon reasonable request.

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
