# Peer review of "RRM2 Regulates Hepatocellular Carcinoma Progression Through Activation of TGF-β/Smad Signaling and Hepatitis B Virus Transcription"

_genes, 2024, doi:10.3390/genes15121575_

Round 1
Reviewer 1 Report
Comments and Suggestions for Authors
The work presented by Wu et al. shows the relevant role of the RRM2 protein in the progression of hepatocellular carcinoma, as well as its role in the background of HBV infection.
In general, article is well presented and is similar to that published by these group for the squalene epoxidase (SQLE) protein and its role in the development of hepatocellular carcinoma in 2022.
There are several important revisions that need to be addressed, and it is also advisable to correct some minor revisions that would be more clarifying.
Major revisions:
- The overexpression of RRM2 in SMMC7721 and Huh2 cell lines is not shown to be significant at the western blot level with respect to LO2. Why the authors have not used HepG2 lines for silencing RRM2? These cell lines seem to have higher RRM2 protein levels.
- To understand the role of RRM2 in EMT it is necessary to investigate the basis for the loss of actin stress fibers in RRM2 silenced cells. It is necessary to show representative bright-field images of these cells and immunofluorescence of actin fibers.
- The relationship described between RRM2 and TGFb is not explained how this interaction occurs in HCC, it would be necessary to establish a direct link such as that performed by Du et al. on the RRM2-YBX1-TGFRB1 axis for pancreatic cancer (Du, Zhouyuan et al. iScience, Volume 27, Issue 10, 110864). It is recommended explore new mechanisms that explain the results obtained either by RNA seq or proteomic study of the silenced RRM2 lines. The data showing that RRM2 participates in smad2 and smad3 activation are not convincing, the correlation values in Figure 4A are less than 0.6. In addition, the western blots for phosphorylation smad2 and smad3 do not show significant results. Perhaps by performing western blots of cells with stable H22 silencing we can see better results.
Minor revision
- The antibody used for RRM2 sc-398294 shows nuclear and cytoplasmic staining but in our case at least there is no nuclear staining of RRM2. Can the authors explain why there is no nuclear staining?
- The sequence used for the control siRNA is not indicated.
- The description of the Real time Quantitative PCR methodology does not describe the detection method (SYBR vs Taqman) or the device where the experiments were performed.
- There is mention that the group has RNA seq data where RRM2 appears overexpressed in HCC but they do not indicate where these data are.
- The IHC images are of poor quality and need to be improved.
- It is not indicated where the RRM2 expression information and the clinical data to generate the Kaplan meier used in the article.
- It would be interesting to study by western blot the levels of RRM2 in the FLAG overexpression assay, not only using the FLAG antibody but also the RRM2 antibody.
- The wound healing studies in Figure 4F show that the negative control does not close the wound at 36 hours. This indicates that this cell line would not be a proper cell line to evaluate in this type of assay.
- Figure 2 and 3 in the Y-axis the word FLOD appears rather of FOLD.
- Figure 5 in IHC header the words p-Samd2/3 appears rather of p-Smad2/3
Author Response
Major revisions:
Comments 1: The overexpression of RRM2 in SMMC7721 and Huh7 cell lines is not shown to be significant at the western blot level with respect to LO2. Why the authors have not used HepG2 lines for silencing RRM2? These cell lines seem to have higher RRM2 protein levels.
Response 1: Thanks for your advice. Compared with the protein level of LO2 cells, RRM2 expression was highest in HepG2.2.15 cells of HCC cell lines. HepG2.2.15 cells are derived from HepG2 with HBV expression stablely [1], and are commonly used to study studies related to HBV infections, and in the present study, we also used HepG2.2.15 cells to conduct a study related to the transcription of HBV. Therefore, in this study, we chose Huh7 cells with relatively high expression as well as SMMC7721 cells with relatively low expression for the study. Thank you again for your suggestions, and we will further consider the selection of cell lines in our subsequent studies to try to make the experimental design more comprehensive.
References:
[1] Sells MA, Chen ML, Acs G. Production of hepatitis B virus particles in Hep G2 cells transfected with cloned hepatitis B virus DNA. Proc Natl Acad Sci U S A. 1987;84(4):1005-1009.
Comments 2: To understand the role of RRM2 in EMT it is necessary to investigate the basis for the loss of actin stress fibers in RRM2 silenced cells. It is necessary to show representative bright-field images of these cells and immunofluorescence of actin fibers.
Response 2: Thanks for your advice. We have supplemented the immunofluorescence experiments of actin fibers (Figure 3E-F) and showed representative bright-field images of these cells (Supplementary Figure S1).
Comments 3: The relationship described between RRM2 and TGFb is not explained how this interaction occurs in HCC, it would be necessary to establish a direct link such as that performed by Du et al. on the RRM2-YBX1-TGFRB1 axis for pancreatic cancer (Du, Zhouyuan et al. iScience, Volume 27, Issue 10, 110864). It is recommended explore new mechanisms that explain the results obtained either by RNA seq or proteomic study of the silenced RRM2 lines. The data showing that RRM2 participates in smad2 and smad3 activation are not convincing, the correlation values in Figure 4A are less than 0.6. In addition, the western blots for phosphorylation smad2 and smad3 do not show significant results. Perhaps by performing western blots of cells with stable H22 silencing we can see better results.
Response 3: Thank you for your valuable advice. Your suggestions will be of great help to us in our further research. We have carefully read the article of Du et al, and in the future, we will construct RRM2-KO HCC cell lines and further explore new mechanisms of RRM2 regulation through research methods such as RNA-seq or proteomics.
Minor revision:
Comments 4: The antibody used for RRM2 sc-398294 shows nuclear and cytoplasmic staining but in our case at least there is no nuclear staining of RRM2. Can the authors explain why there is no nuclear staining?
Response 4: Thanks for your reminding. We considered that the absence of nuclear staining by the RRM2 antibody could be due to the difference in tissue types, the literatures show that RRM2 in liver tissue is mainly distributed in the cytoplasm [1-2], which is consistent with the results of this study. In addition, we also used RRM2 antibody (Proteintech, 11661-1-AP), and the results also showed that RRM2 in liver tissues was mainly stained in the cytoplasm.
References:
[1] Zhang X, Qi M, Huo K, et al. Celastrol induces ferroptosis by suppressing RRM2 in hepatocellular carcinoma. Heliyon. 2024;10(13):e33936.
[2] Lee B, Ha SY, Song DH, Lee HW, Cho SY, Park CK. High expression of ribonucleotide reductase subunit M2 correlates with poor prognosis of hepatocellular carcinoma. Gut Liver. 2014;8(6):662-668.
Comments 5: The sequence used for the control siRNA is not indicated.
Response 5: Thank you for your reminding. We have added the sequence used for the control siRNA in the manuscript.
Comments 6: The description of the Real time Quantitative PCR methodology does not describe the detection method (SYBR vs Taqman) or the device where the experiments were performed.
Response 6: Thank you for your reminding. We have provided a more detailed description of the Real time Quantitative PCR methodology in the manuscript.
Comments 7: There is mention that the group has RNA seq data where RRM2 appears overexpressed in HCC but they do not indicate where these data are.
Response 7: Thank you for your suggestion. The results of RNA-seq data analysis of clinical samples of hepatocellular carcinoma done by our group have been published in the article cited in the text, and according to your comments, we have displayed this part of the analysis on the high expression of RRM2 in hepatocellular carcinoma in the manuscript.
Comments 8: The IHC images are of poor quality and need to be improved.
Response 8: Thank you for your suggestion. We have re-uploaded IHC images, hoping to improve the quality of the picture.
Comments 9: It is not indicated where the RRM2 expression information and the clinical data to generate the Kaplan meier used in the article.
Response 9: Thank you for your reminding. RRM2 expression information and clinical data we analyzed directly on the Kaplan-Meier Plotter website, whose URL we have added to the manuscript.
Comments 10: It would be interesting to study by western blot the levels of RRM2 in the FLAG overexpression assay, not only using the FLAG antibody but also the RRM2 antibody.
Response 10: Thanks for your advice. We have used both FLAG and RRM2 antibodies in Western blotting to study the level of RRM2 in FLAG overexpression assay (Figure 1G).
Comments 11: The wound healing studies in Figure 4F show that the negative control does not close the wound at 36 hours. This indicates that this cell line would not be a proper cell line to evaluate in this type of assay.
Response 11: Thanks for your advice. The wound healing study is a common assay for detecting the migration of tumor cells, and several papers used the SMMC7721 or Huh7 cells to perform wound healing assay to assess the migration ability of HCC cells [1-5]. In addition, the transwell assay and the F-actin fluorescence assay you suggested earlier were also used in this paper to further detect the migratory ability of HCC cells.
References:
[1] Dai M, Chen N, Li J, et al. In vitro and in vivo anti-metastatic effect of the alkaliod matrine from Sophora flavecens on hepatocellular carcinoma and its mechanisms. Phytomedicine. 2021;87:153580.
[2] Khan H, Ni Z, Feng H, et al. Combination of curcumin with N-n-butyl haloperidol iodide inhibits hepatocellular carcinoma malignant proliferation by downregulating enhancer of zeste homolog 2 (EZH2) - lncRNA H19 to silence Wnt/β-catenin signaling. Phytomedicine. 2021;91:153706.
[3] Shan M, Liu D, Sun L, et al. KIAA1429 facilitates metastasis via m6A-YTHDC1-dependent RND3 down-regulation in hepatocellular carcinoma cells. Cancer Lett. 2024;584:216598.
[4] Xiao T, Bao J, Tian J, et al. Flavokawain A suppresses the vasculogenic mimicry of HCC by inhibiting CXCL12 mediated EMT. Phytomedicine. 2023;112:154687.
[5] Shan M, Liu D, Sun L, et al. KIAA1429 facilitates metastasis via m6A-YTHDC1-dependent RND3 down-regulation in hepatocellular carcinoma cells. Cancer Lett. 2024;584:216598.
Comments 12: Figure 2 and 3 in the Y-axis the word FLOD appears rather of FOLD.
Response 12: Thanks for your advice. We have made revisions in the manuscript.
Comments 13: Figure 5 in IHC header the words p-Samd2/3 appears rather of p-Smad2/3
Response 13: Thanks for your advice. We have made revisions in the manuscript.
On behalf of all the contributing authors, I would like to express our sincere appreciations of your constructive comments. We are glad to have such a professional and responsible reviewer as you, and we really appreciate your help and patience. We are deeply aware of the shortcomings of our own study, which will become the driving force for us to continuously improve our own scientific research quality and scientific research level. We hope that the changes we made to the manuscript will be satisfactory to you. Thanks a lot.

Reviewer 2 Report
Comments and Suggestions for Authors
A well rounded job describing the role of RRM2 in hepatocrllular carcinoma.
Minor comments: In the abdtract the conclusion section is not supported by the results.
Introduction: I believe that the authors should not really mentioned the role of TGFβ and of BMP.
Methods : The authors should explain the methodology clearer in some instances ( JetPRIME, EdU assay).
Author Response
Comments 1: In the abstract the conclusion section is not supported by the results.
Response 1: Thank you for your rigorous and precise attitude. We have revised the abstract section of the manuscript.
Comments 2: I believe that the authors should not really mentioned the role of TGFβ and of BMP.
Response 2: Thank you for your kind suggestion. We carefully read the literature related to TGFβ and BMP and made some changes to the manuscript, with relevant content highlighted in red.
Comments 3: The authors should explain the methodology clearer in some instances (JetPRIME, EdU assay).
Response 3: Thanks for your advice. We have reworked the description of the detection process for experimental methods such as JetPRIME and EdU.
Overall, we are especially grateful to you for your careful review of this manuscript and your valuable comments. We are well aware that our work is not perfect enough, and this will also be the driving force of our future research. Thanks again.

Reviewer 3 Report
Comments and Suggestions for Authors
In their manuscript, Wu et al focus on the role of RRM2 in hepatocellular carcinoma and suggest that its pro-oncogenic role in HCC is mediated through regulating the TGFbeta signaling pathway and the transcription of hepatitis B virus. I have the following comments for improvement of the current study:
1) Since RRM2 positively regulates both cell proliferation and migration, I wonder how did the authors perform their wound healing assays, to demostrate that RRM2 indeed affects migration. In the method description, it is not mentioned whether for example a cytostatic compound, such as mitomycin has been used to inhibit proliferation during the wound healing assay. If not, I recommend repeating these experiments in the presence of this proliferation inhibitor.
2) In the expression data retrieved from GEPIA, RRM2 positively correlates to SMAD2 and SMAD3 expression, however in the in vitro experiments, when RRM2 is silenced, SMAD2 or SMAD3 protein expression does not change. How do the authors explain this discrepancy? Also, what is the correlation between RRM2 and TGFB1 expression levels in tumors (same cohort from GEPIA)?
3) Since RRM2 greatly affects TGFbeta signaling, the expression (both mRNA and protein) of typical TGFbeta-target genes upon RRM2 silencing or over-expression should be evaluated (such as PAI-1, SMAD7 etc).
4) Have the authors investigated whether TGFbeta regulates RRM2 expression in HCC cell lines?
Comments on the Quality of English LanguageThe overall quality of english language is good, however the authors should double check for grammatical errors throughout the text.
Author Response
Comments 1: Since RRM2 positively regulates both cell proliferation and migration, I wonder how did the authors perform their wound healing assays, to demostrate that RRM2 indeed affects migration. In the method description, it is not mentioned whether for example a cytostatic compound, such as mitomycin has been used to inhibit proliferation during the wound healing assay. If not, I recommend repeating these experiments in the presence of this proliferation inhibitor.
Response 1: Thanks for your advice. In the wound healing assay, when linear scratches were prepared, we continued the culture using fresh serum-free medium, which inhibited cell proliferation to some extent. in addition, we also demonstrated that RRM2 did affect the migration of HCC cells by a variety of methods such as transwell assay, as well as EMT and the newly supplemented F-actin fluordscence assay. The suggestion of performing wound healing assays in the presence of a proliferation inhibitor is great and will be helpful for our subsequent studies. Thank you again for your suggestion.
Comments 2: In the expression data retrieved from GEPIA, RRM2 positively correlates to SMAD2 and SMAD3 expression, however in the in vitro experiments, when RRM2 is silenced, SMAD2 or SMAD3 protein expression does not change. How do the authors explain this discrepancy? Also, what is the correlation between RRM2 and TGFB1 expression levels in tumors (same cohort from GEPIA)?
Response 2: Thank you very much for reviewing our manuscript. We found that RRM2 had some correlation with SMAD2 and SMAD3, but the correlation coefficients were not high, both lower than 0.6. All, we examined the phosphorylation levels of SMAD2 and SMAD3, and found that RRM2 could significantly up-regulate their phosphorylation levels, although it had no obvious effect on the protein expression of SMAD2 and SMAD3. In addition, we did not retrieve the results of RRM2 correlation with TGFB1 expression in the GEPIA database, but we found that RRM2 could promote the expression of TGFB1 by Western blotting and Q-PCR experiments. Recent studies have also found that RRM2 promotes liver metastasis of pancreatic cancer by stabilizing YBX1 and activating the TGF-beta pathway [1].
References:
[1] Du Z, Zhang Q, Xiang X, et al. RRM2 promotes liver metastasis of pancreatic cancer by stabilizing YBX1 and activating the TGF-beta pathway. iScience. 2024;27(10):110864.
Comments 3: Since RRM2 greatly affects TGFbeta signaling, the expression (both mRNA and protein) of typical TGFbeta-target genes upon RRM2 silencing or over-expression should be evaluated (such as PAI-1, SMAD7 etc).
Response 3: Thank you for your valuable and thoughtful comments. SMAD2/3 are two important downstream genes of the TGF-β signaling pathway that are associated with liver fibrosis and carcinogenesis, and through database, in vitro and in vivo experiments, we have confirmed that RRM2 can regulate the phosphorylation level of SAMD2 and SAMD3, and our preliminary study has clarified that RRM2 can regulate TGF-β and its downstream activation of SAMD2 and SAMD3. PAI-1, SMAD7 are also important target proteins of TGF-β, but due to time constraints, we have not carried out further studies on them. We plan to construct RRM2-KO HCC cell lines and perform RNA-seq and proteomics related assays to further explore the regulation of TGF-β and its downstream target proteins by RRM2. Thank you again for your professional advice and guidance.
Comments 4: Have the authors investigated whether TGFbeta regulates RRM2 expression in HCC cell lines?
Response 4: Thanks for your advice. Our previous work focused on the regulatory effect of RRM2 on TGF-β, and the effect of TGF-β on RRM2 expression levels in HCC cell lines has not been examined. Your comments are very helpful to us, and we will further explore the regulatory role between RRM2 and TGF-β in subsequent studies, and further elaborate on the role of RRM2 in the progression of HCC and its related mechanism.
Comments on the Quality of English Language: The overall quality of English language is good, however the authors should double check for grammatical errors throughout the text.
Response: Thank you for your reminding. We have carefully revised the grammar throughout the manuscript.
We feel great thanks for your professional review work on our article. In this process, I see my own shortcomings and understand the direction that I need to improve. It was a great benefit.

Round 2
Reviewer 3 Report
Comments and Suggestions for Authors
The authors basically did not respond to any of my comments by performing a few simple experiments or by simply incorporating expression correlation graphs from GEPIA. Therefore, I reject the paper.
Comments on the Quality of English LanguageThe quality of English language can be improved
Author Response
Comments 1: The authors basically did not respond to any of my comments by performing a few simple experiments or by simply incorporating expression correlation graphs from GEPIA. Therefore, I reject the paper.
Response 1: We apologize for not providing a reasonable response to your previous comments(“In the expression data retrieved from GEPIA, RRM2 positively correlates to SMAD2 and SMAD3 expression, however in the in vitro experiments, when RRM2 is silenced, SMAD2 or SMAD3 protein expression does not change. How do the authors explain this discrepancy? Also, what is the correlation between RRM2 and TGFB1 expression levels in tumors? ”). We carefully reviewed the relevant literature and performed relevant experiments and reanalyzed the GEPIA database. In this study, we analyzed the GEPIA database and found that RRM2 was correlated with SMAD or SMAD3. Supplementary Q-PCR experiments also revealed that knockdown of RRM2 decreased the mRNA expression levels of SMAD2 and SMAD3, while overexpression of RRM2 upregulated the mRNA expression levels of SMAD2 and SMAD3 (Supplementary Figure S2). The results of previous WB experiments revealed that RRM2 had no significant effect on the protein expression levels of SMAD2 and SMAD3, but RRM2 was able to upregulate the protein phosphorylation levels of SMAD2 and SMAD3. To explain this discrepancy we reviewed the relevant literature, and It has been shown that phosphorylation of SMAD2 or SMAD3 is followed by ubiquitination and subsequent degradation by the proteasome [1-5]. Therefore, we hypothesized that RRM2 could upregulate mRNA expression, and theoretically the protein level of SMAD2 or SMAD3 should also increase, but the total protein levels did not change significantly due to partial degradation of phosphorylated SMAD2 or SMAD3 proteins. Regarding this part we have also added it to the discussion section of the manuscript and labeled it with red fonts, and we will subsequently design experiments for further in-depth study and validation. In addition, we also reanalyzed the GEPIA database and found that the expression levels of RRM2 and TGFB1 were also correlated (Figure 4A), and this part of the results was also added in the manuscript. Thank you again for your comments, which made me deeply realize our shortcomings, and I am very sorry that I did not give a reasonable reply before, and I hope you will be satisfied with this reply.
References:
[1] Lo RS, Massagué J. Ubiquitin-dependent degradation of TGF-beta-activated smad2. Nat Cell Biol. 1999;1(8):472-478.
[2] Yu JS, Ramasamy TS, Murphy N, et al. PI3K/mTORC2 regulates TGF-β/Activin signalling by modulating Smad2/3 activity via linker phosphorylation. Nat Commun. 2015;6:7212.
[3] Herhaus L, Al-Salihi M, Macartney T, Weidlich S, Sapkota GP. OTUB1 enhances TGFβ signalling by inhibiting the ubiquitylation and degradation of active SMAD2/3. Nat Commun. 2013;4:2519.
[4] Gao S, Alarcón C, Sapkota G, et al. Ubiquitin ligase Nedd4L targets activated Smad2/3 to limit TGF-beta signaling. Mol Cell. 2009;36(3):457-468.
[5] Wang C, Li Y, Zhang H, et al. Oncogenic PAK4 regulates Smad2/3 axis involving gastric tumorigenesis. Oncogene. 2014;33(26):3473-3484.
Comments on the Quality of English Language: The quality of English language can be improved.
Response: Thanks again. We have carefully checked and improved the English writing in the revised manuscript.
On behalf of all contributing authors, I would like to apologize for not being able to give a reasonable explanation for your previous comments. We are glad to have such a professional and responsible reviewer as you, and we sincerely thank you for your constructive comments. We are deeply aware of the shortcomings of our own research, which will be our motivation to continuously improve the quality and level of our own research. We hope that our revision of the manuscript will satisfy you. Thank you very much.

Round 3
Reviewer 3 Report
Comments and Suggestions for Authors
Dear authors,
Thank you for providing new data during this round of revision. I have just a couple of comments. First, your interpretation that the SMAD protein levels do not change because of a possible phosphorylation-induced degradation of the proteins are not correct because this could be true in the case of SMAD linker phosphorylation. The SMAD linker phosphorylation is mediated by kinases but not by the TGFbeta receptor, which activates the pathway. I suppose that you checked into the C-terminal phosphorylation that is associated with the activation of SMADs and downstream signaling. However, the authors should provide a table with the antibodies used in this study as, I was not able to evaluate which antibodies you have used. This is a very important information that allows other researchers in the future to be able to reproduce the results.
My second request is if you can provide raw data (Ct values, relative expression etc) for the RT-qPCR shown (for example for the new data showing mRNA expression of PAI-1 and SMAD7, but also for previous RT-qPCRs)
Author Response
Comments 1: Thank you for providing new data during this round of revision. I have just a couple of comments. First, your interpretation that the SMAD protein levels do not change because of a possible phosphorylation-induced degradation of the proteins are not correct because this could be true in the case of SMAD linker phosphorylation. The SMAD linker phosphorylation is mediated by kinases but not by the TGFbeta receptor, which activates the pathway. I suppose that you checked into the C-terminal phosphorylation that is associated with the activation of SMADs and downstream signaling. However, the authors should provide a table with the antibodies used in this study as, I was not able to evaluate which antibodies you have used. This is a very important information that allows other researchers in the future to be able to reproduce the results.
Response 1: Thank you very much for your suggestion, we have listed the antibodies used in this study in the Materials section of the manuscript and highlighted them in red font. Secondly, your reference to the effect of RRM2 on SMAD protein levels in the presence of SMAD linker phosphorylation may be true, and we very much recognize this idea and have made the relevant changes in the Discussion section. We will also subsequently design experiments to further investigate this hypothesis.
Comments 2: My second request is if you can provide raw data (Ct values, relative expression etc) for the RT-qPCR shown (for example for the new data showing mRNA expression of PAI-1 and SMAD7, but also for previous RT-qPCRs)
Response 2: Thank you very much for your suggestion, we have compiled the raw data (Ct value, relative expression, etc.) of RT-qPCR in this study and listed them in the Supplementary Material.
On behalf of all the contributing authors, I would like to express our sincere appreciations of your constructive comments. We are glad to have such a professional and responsible reviewer as you, and we really appreciate your help and patience. We are deeply aware of the shortcomings of our own study, which will become the driving force for us to continuously improve our own scientific research quality and scientific research level. We hope that the changes we made to the manuscript will be satisfactory to you. Thanks a lot.